# Sustainable Development Goal Attainment in the Wake of COVID-19: Simulating an Ambitious Policy Push

Taylor Hanna [1], Barry B. Hughes [1], Mohammod T. Irfan [1], David K. Bohl [1], José Solórzano [1], Babatunde Abidoye [2], Laurel Patterson [2] and Jonathan D. Moyer [1,*]

[1]  Frederick S. Pardee Center for International Futures, Josef Korbel School of International Studies, University of Denver, 2201 S. Gaylord St., Denver, CO 80208, USA; taylor.hanna@du.edu (T.H.); barry.hughes@du.edu (B.B.H.); mohammod.irfan@du.edu (M.T.I.); davidkbohl@gmail.com (D.K.B.); jose.solorzano@du.edu (J.S.)
[2]  United Nations Development Programme, 1 United Nations Plaza, New York, NY 10017, USA; babatunde.abidoye@undp.org (B.A.); laurel.patterson@undp.org (L.P.)
*   Correspondence: jmoyer@du.edu

**Abstract:** Even before the COVID-19 pandemic, the world was not on course to meet key Sustainable Development Goals (SDGs) including SDG 1 (No Poverty) and SDG 2 (Zero Hunger). Some significant degree of additional effort was needed before the pandemic, and the challenge is now greater. Analyzing the prospects for meeting these goals requires attention to the combined effects of the pandemic and such additional impetus. This article assesses the impact of the COVID-19 pandemic on progress toward the SDGs and explores strategies to recover and accelerate development. Utilizing the International Futures (IFs) forecasting system and recognizing the near impossibility of meeting the goals by 2030, three scenarios are examined through to 2050: A pre-COVID-19 trajectory (*No COVID-19*), the current path influenced by the pandemic (*Current Path*), and a transformative SDG-focused approach prioritizing key policy strategies to accelerate outcomes (*SDG Push*). The pandemic led to a rise in extreme poverty and hunger, with recovery projected to be slow. The *SDG Push* scenario effectively addresses this, surpassing the *Current Path* and achieving significant global improvements in poverty, malnutrition, and human development by 2050 even relative to the *No COVID-19* path. The findings emphasize the need for integrated, transformative actions to propel sustainable development.

**Keywords:** COVID-19; sustainable development goals; human development; poverty; hunger; forecasting human development



## 1. Introduction

The Sustainable Development Goals (SDGs) serve as the global framework for driving progress toward accelerating human development. Even at their inception, the SDGs were understood as highly ambitious, and progress since 2015 has not been on pace to achieve the goals by the target date of 2030 [1]. In 2020, the outbreak of the COVID-19 pandemic led to not only more than 1.8 million deaths [2] but also shutdowns and mitigation measures worldwide, slowing economic growth. Inequality increased both across and within developing countries [3,4], and for the first time in decades, the global poverty rate increased, signaling a reversal of recent progress. While the economy has rebounded, growth has settled back to a positive but moderate pace. This along with the many other effects of COVID-19 is expected to have implications for progress toward SDG achievement across the SDG agenda [5–7].

Previous work explored how COVID-19 would impact the SDGs, with some studies broadly assessing the literature on developmental outcomes [5,8] or focusing on short-term quantitative outcomes across a wide range of indicators [9]. This research finds that the pandemic has indeed negatively affected many SDGs [8,10], setting overall progress back 8.2 percent in its first year [6]. Other research has focused on specific SDG indicators,

including global poverty [3,11–16], food affordability [17], food insecurity [14], hunger [18], and maternal and child health and undernutrition [19,20]. This literature is also focused on effects in the first few years after COVID-19. Cooper et al. [21] project moderate and severe food insecurity through to 2030, though not in comparison to a baseline without the pandemic. Other work projects the outcomes from an integrated development push on SDG achievement but without considering the effect of COVID-19 [22].

Now, more than halfway through the SDG horizon and several years out of the initial COVID-19 outbreak, we fill gaps in the literature by reassessing progress and taking stock of the path we are currently on while assessing prospects for accelerating development. This paper advances the understanding of how the COVID-19 pandemic has affected progress toward achieving the first two SDGs (SDG 1: No Poverty; SDG 2: Zero Hunger) and improving the Human Development Index (HDI), which summarizes progress towards several of the SDGs.

It then turns to explore how an ambitious push toward global development might make up for that setback. Previous work has proposed strategies for global responses that push beyond addressing the pandemic's immediate effects [23], ranking response strategies in relation to the SDGs [24], and even mapped out potential future scenarios and how SDG progress may be positively or negatively affected [25]. There remains a gap in the literature related to quantitatively assessing responses to the generally accepted developmental setback caused by COVID-19 and how these responses might alter development in the long run. We fill this gap by exploring alternative multidimensional policy strategies that can improve long-term human wellbeing in spite of the pandemic.

We find that even prior to the COVID-19 outbreak, the world was not on track to achieve the SDG agenda, reinforcing findings in the literature. We find that the effect of COVID-19 had an immediate adverse impact on progress toward the SDGs and that the shock will cast a long shadow, setting back progress for decades from where it would have otherwise been. In addition, we find that significant improvements can be made to the development trajectory that improve SDG attainment if a set of policy priorities are pursued.

COVID-19 will not be the last shock to challenge global development, and it will be important to better understand how to overcome both it and challenges in the future. Therefore, one of our scenarios explores how a substantial and transformative agenda could accelerate progress toward the SDG targets by midcentury. This offers insight into how integrated action across policy areas can further sustainable development and set the world on a new path forward. We find that this scenario, while it does not result in achieving the Goals on time in all countries, is successful in quickly making up for the damage inflicted by COVID-19 and further in propelling progress for decades to come.

This paper proceeds by first elaborating the methods used to drive the analysis with a particular emphasis on the structure of the modeling framework and scenario assumptions. Next, we present the results, highlighting how the COVID-19 pandemic is likely to change long-term development outcomes and what a successful set of policy strategies can do to further improve development outcomes beyond our *Current Path* of development. Finally, we discuss these findings and highlight methodological challenges as well as some implications of the policy strategies that are modeled.

## 2. Materials and Methods

### 2.1. International Futures

This study uses the International Futures (IFs) forecasting system for forecasting and scenario analysis. IFs is an integrated assessment modeling platform with representation of 188 countries and capability to forecast out to 2100. It features numerous endogenized and interconnected sub-models with coverage of the following systems: agriculture [26], economics, education [27], energy, environment [28], demographics, governance [29,30], health [31], infrastructure [32], international politics [33], and technology. IFs is open-source and is free to use online or to download for offline use. The following sections describe key

areas of the model for this work, but extensive model information and documentation is available for further detail [34].

IFs forecasts patterns of long-term economic growth using a recursive dynamic general equilibrium-seeking structure with a Cobb–Douglas production function and an endogenously dynamic Solow residual. Six capital sectors, labor by skill level, and the endogenously driven productivity term are shaped by forces within each of the 188 countries but also by international trade, official development assistance, foreign direct investment flows, and international migration with associated remittance patterns. A social accounting matrix structure accounts for flows across economic sectors and between households, firms, and governments and with the rest of the world. Representation of government finance within the social accounting matrix includes specification of revenues from domestic taxation streams and foreign assistance, while identified expenditures include transfer payments and direct spending in military, health, education, research and development (R&D), infrastructure, and residual categories. Other core features of the IFs model include partial-equilibrium agriculture and energy models physically elaborating those sectors of the total economy, an infrastructure model with access to information and communication technologies, electricity, water and sanitation, and paved roads [32], as well as well as an education model which simulates the grade-level flow of students through the education system and the pattern of educational attainment across adult life spans.

Although the IFs system forecasts variables related to selected targets of all SDG goals, we focus here on three core outcome indicators in IFs: poverty, undernutrition, and the HDI, each described below.

### 2.1.1. Poverty

Poverty rates in IFs are initialized using data from the World Bank [35], which come originally from household surveys and are driven by the model's economic growth and inequality models [36]. The social accounting matrix structure of the economic model tracks financial flows to and from households resulting from labor earnings and transfers in interaction with firms and governments. Resultant disposable income is allocated to consumption or savings based on long-term country development patterns, demographic structure, and sectoral prices and interest rates. Poverty rates are estimated at per-capita household consumption levels assuming a log-normal distribution of household income, the shape of which is affected by changes to the Gini coefficient.

For this analysis, we focus on extreme poverty, using the recently updated international poverty line of USD 2.15/day in 2017 US dollars at purchasing power parity. For countries lacking data values, the model estimates initial values using a cross-sectional function with GDP per capita.

### 2.1.2. Malnutrition

Building on variables from and interactions across the demographic, general equilibrium economic, and partial-equilibrium agriculture models, IFs forecasts the prevalence of malnutrition as a function of available calories per capita, a coefficient of variation, and a minimum dietary energy requirement. The partial-equilibrium agriculture sub-model represents crop, meat, and fish production and trade and therefore calorie and protein availability [26], while the economic model generates consumption potential and calorie demand per capita as functions of GDP per capita and food prices. Demographics shape total country demand levels. As with the income that shapes poverty levels, access to calories is assumed to be distributed log-normally. The shape of the distribution is determined by the caloric coefficient of variation, which is driven by income growth, inequality, and social inequality, as represented by female labor participation and the youth dependency ratio. Data from the FAO are used to initialize the prevalence of malnutrition as well as calories per capita, the coefficient of variation, and the minimum daily energy requirement [26].

This analysis is focused on population-wide malnutrition to provide the broadest picture of progress toward eliminating hunger. It does not account for differing levels of hunger by gender or for young children, measures of which are also available in IFs.

### 2.1.3. Human Development Index

The HDI has been designed and maintained by the United Nations Development Programme (UNDP) to measure general levels of human development in all countries across three basic dimensions—health, education, and living standards. The UNDP replaced an earlier and simpler version of the HDI with a more refined version in 2010 [37]. This index is a geometric mean of three normalized sub-indices, representing (1) life expectancy at birth, (2) an average of mean years of schooling completed at age 25 or older and the expected years of schooling upon entry to education, and (3) a logarithm of gross national income per capita at purchasing power parity (for which IFs substitutes gross domestic product at PPP). Data from the UNDP initialize the HDI values for each country in the 2019 base year of IFs [35].

Forecasts of the HDI in IFs are driven by several sub-models, especially the demographic, health, education, and economic models. The health model produces the life expectancy index. This model, drawing on data and approaches of the Global Burden of Disease project [38], represents 15 causes of age- and sex-specific mortality across communicable, non-communicable, and accident and injury categories, thereby providing the basis for the computation of life expectancy. The education model represents year-specific entry into and flow through primary, lower-secondary, upper-secondary, and tertiary education; the progression through these levels feeds the years of schooling attained by population cohorts at post-educational ages, and the demographic model carries age-specific education through the variable life spans of cohort members. The economic and demographic models determine GDP at PPP.

### 2.2. Scenarios

We explore three scenarios aimed at evaluating where we are in terms of progress toward the SDGs and how we can collectively begin to narrow the gap between the road we are on and one that achieves the SDG agenda. These scenarios are modified from a set of four scenarios originally produced in 2020 and 2021 [39]. They have since been updated and modified to reflect recent data and the literature and run in an updated version of the IFs model. See Table 1 and following sections for a brief description of the three scenarios. Due to significant volatility in growth projections, Venezuela has been removed from the country set for this study.

**Table 1.** Description of scenarios used in this analysis.

| Scenario Name | Description |
|---|---|
| *No COVID-19* | This scenario is a projection of the development path that the world was on prior to the COVID-19 outbreak. |
| *Current Path* | The *Current Path* reflects a baseline path of development in the future, including the effect of COVID-19. |
| *SDG Push* | This scenario simulates an integrated push toward SDG achievement through ambitious but achievable global interventions. |

### 2.2.1. Current Path

The *Current Path* can be thought of as the baseline development path, with the impacts of COVID-19 but without additional major shocks and without transformative policy change. Using the interconnected sub-models in IFs, this scenario reflects a dynamic unfolding of development patterns within and across countries as well as sectors. The *Current Path* uses exogenously imposed GDP growth rate data and projections from the latest version of the IMF's World Economic Outlook through to 2025 [40]. The *Current Path* in IFs has been

used widely in academic and policy-oriented work to describe the path that the world is currently on [28,30,33].

### 2.2.2. No COVID-19

The *No COVID-19* scenario serves as a counterfactual, simulating the path we would be on had there been no COVID-19 outbreak, and thus allows us to make a rough assessment of the pandemic impact on the goal path. It leans on the same logic that informs the *Current Path* scenario but uses data and projections made prior to the outbreak of COVID-19. GDP growth rate projections from the IMF World Economic Report released in October 2019 [41] are imposed exogenously through to 2025.

### 2.2.3. SDG Push

In the wake of the pandemic, UNDP [23] put forth guidance on how the world might not only recover from the pandemic but move beyond recovery to accelerate progress toward the SDGs, defining four key areas of response: governance (building a new social contract), social protection (uprooting inequalities), green economy (rebalancing nature, climate, economy), and digital disruption and innovation (for speed and scale). This *SDG Push* scenario is based on initial work by Hughes et al. [39], oriented around the key areas outlined by UNDP, and further builds on work by Moyer and Hedden [42]. Specific details about the individual scenario interventions and parameter changes within IFs are available in the Supplementary Information.

In this scenario, the world pursues a set of policies that are designed to further sustainable development within planetary boundaries. Beginning with agricultural systems, sustainable development transformations include a shift away from meat-based diets towards plant-based diets, an increase in agricultural yields, and a reduction in loss (including losses in production, transmission, and consumption). In addition to the resulting increase in caloric availability, we also assume an increase in the equity of the distribution of calories, a simulation of cash transfer or food subsidy programs. Governments increasingly focus on programs that are core to human development, boosting spending on infrastructure, education, health (including a focus on family planning), and R&D while also increasing household transfers for welfare and pensions. Households benefit from expanded access to safe water, sanitation, information communication technology, and access to electricity as well as a reduction in traditional cookstoves. While government spending is important, governments in this scenario also improve the efficacy of this spending along with increasingly democratic institutions.

The scenario also simulates a transformation in energy systems by implementing a progressive carbon tax (to USD 200 per ton for OECD countries and USD 50 for non-OECD countries), a progressive reduction in energy demand (greater and more rapid in OECD countries than in non-OECD countries), and improvements in energy efficiency. Future coal production is constrained while renewable energy development and investment is accelerated. Further environmental policies reduce overall water demand relative to the *Current Path*, reduce urban air pollutants, and increase forested land.

The cumulative effect of these interventions is to make development less carbon intensive, more efficient, and less wasteful, while also pointing resources towards areas of investment that are crucial to multidimensional human wellbeing.

## 3. Results

The following sections include the results of all three scenarios across three key outcome indicators: the population in extreme poverty, the undernourished population, and the Human Development Index.

### 3.1. Poverty

The elimination of poverty is the first SDG (SDG 1) and highly connected to many of the others. Here, we focus on the international extreme poverty line of USD 2.15/day using

2017 US dollars at purchasing power parity. For this analysis, a country or region is said to have eliminated extreme poverty if the portion of the population living below the extreme poverty line falls below 3 percent. Full global results are available in Table 2.

**Table 2.** Results by scenario for SDG 1.1, using the percent of the population living on less than USD 2.15/day in 2017 US dollars. Source: IFs 8.10.

| Scenario | 2019 Global Value | 2019 Countries Meeting Target | 2030 Global Value | 2030 Countries Meeting Target | 2050 Global Value | 2050 Countries Meeting Target |
|---|---|---|---|---|---|---|
| *No COVID-19* | 9 | 102 | 6.8 | 113 | 2.8 | 140 |
| *Current Path* | 9 | 102 | 7.5 | 107 | 3.1 | 139 |
| *SDG Push* | 9 | 102 | 6.6 | 117 | 1.5 | 159 |

Even prior to the outbreak of COVID-19, the world was not on track to meet SDG 1. Globally, an estimated 9 percent of the population (798 million people) lived in extreme poverty. Along the *No COVID-19* trajectory, poverty was expected to decline gradually. In this scenario, 6.6 percent of the population (578 million) would still live in extreme poverty in 2030. By 2050, the world at a global level just meets the target, with 2.8 percent of the world (269 million). At a country level, 102 countries are estimated to have already met the SDG target in 2019. By 2050, they would be joined by an additional 37 countries achieving the goal.

Along the *Current Path* affected by COVID-19, slowed economic growth resulted in an increase in poverty that could continue to affect progress toward SDG 1 for some time. In 2020 alone, we estimate an increase in the extreme poverty rate of 1 percentage point, reflecting nearly 80 million people pushed into extreme poverty by the pandemic in that year. This is a slightly greater effect than seen in previous work estimating the effect of COVID-19 on poverty using the IFs model (an estimated 73.9 million) [43] and the World Bank (estimating over 70 million) [44] and somewhat less than an estimates by Laborde et al. [14] in the most recent work by Mahler et al. [3], which finds a COVID-19-induced increase in extreme poverty of 1.2 percentage points (90 million). As the economy rebounded somewhat, we forecast poverty reductions after the initial year, but these improvements will remain slow and behind the *No COVID-19* counterfactual. By 2030, we project 7.5 percent of the population (635 million) in extreme poverty, still nearly 57 million more than the *No COVID-19* scenario in the same year (Figure 1). By 2050, the world just misses reaching the target, with 3.1 percent of the population (298 million) still in poverty.

In the *SDG Push*, poverty reduction accelerates as a result of interventions which boost growth and sustainable development. The poverty rate in the *SDG Push* scenario falls below that in the *No COVID-19* world by 2029. By 2030, the extreme poverty rate reaches 6.6 percent (499 million people in extreme poverty, which is 81 million fewer than in the *Current Path* headcount). Global extreme poverty falls below 3 percent by 2042, and by 2050, it falls to 1.5 percent (104 million, or 137 million fewer than in the *Current Path*). Along the *Current Path*, the global poverty rate is projected to remain above 3 percent through the horizon chosen for this analysis.

At a regional level, sub-Saharan Africa is home to the most people living in extreme poverty, with an estimated 404 million in 2019. But the effect of COVID-19 in the region was not as severe as in Central and Southern Asia, where 46 million people were pushed into extreme poverty due to COVID-19 in 2020, compared with just under 20 million in SSA (Figure 2). However, in the following years, the poverty difference in the CSA region is expected to fall, while in the SSA region it remains relatively steady, reflecting faster population growth in the sub-Saharan Africa region.

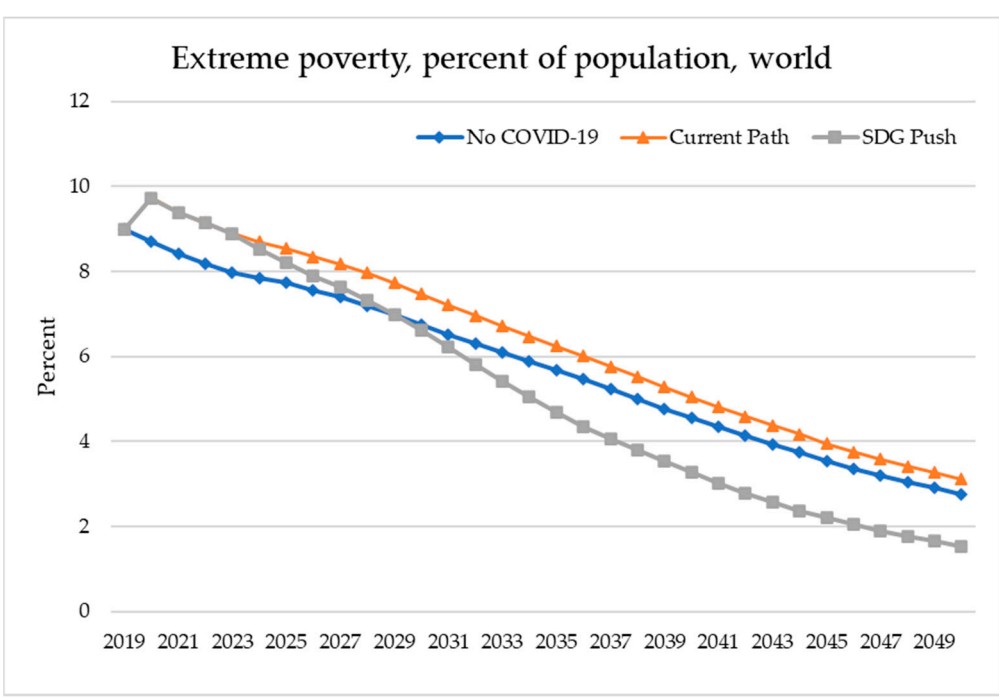

**Figure 1.** Percent of world population living on less than USD 2.15/day across scenarios. Source: IFs 8.10.

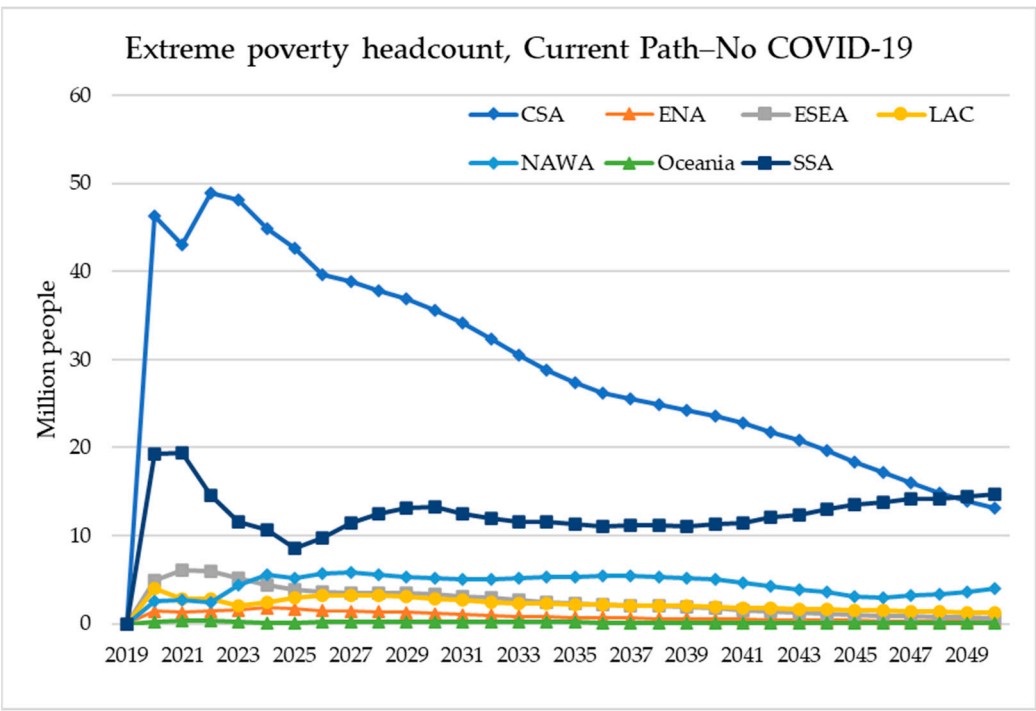

**Figure 2.** Difference between the number of people in poverty in the *Current Path* scenario and the *No COVID-19* scenario, by region. Source: IFs 8.10.

Figure 3 shows the rate of extreme poverty by region across all three scenarios in 2030 and 2050. The *SDG Push* begins to improve extreme poverty in regions where it is the most prevalent relative to the *Current Path*. In Europe and Northern America (ENA), Latin America and the Caribbean (LAC), and sub-Saharan Africa (SSA), the *SDG Push* makes up for the difference between the *Current Path* and *No COVID-19* scenarios by 2030, while in others—Central and Southern Asia (CSA), Eastern and South-Eastern Asia (ESEA),

Northern Africa and Western Asia (NAWA), and Oceania—the *SDG Push* still lags behind the *No COVID-19* scenario (Figure 3a).

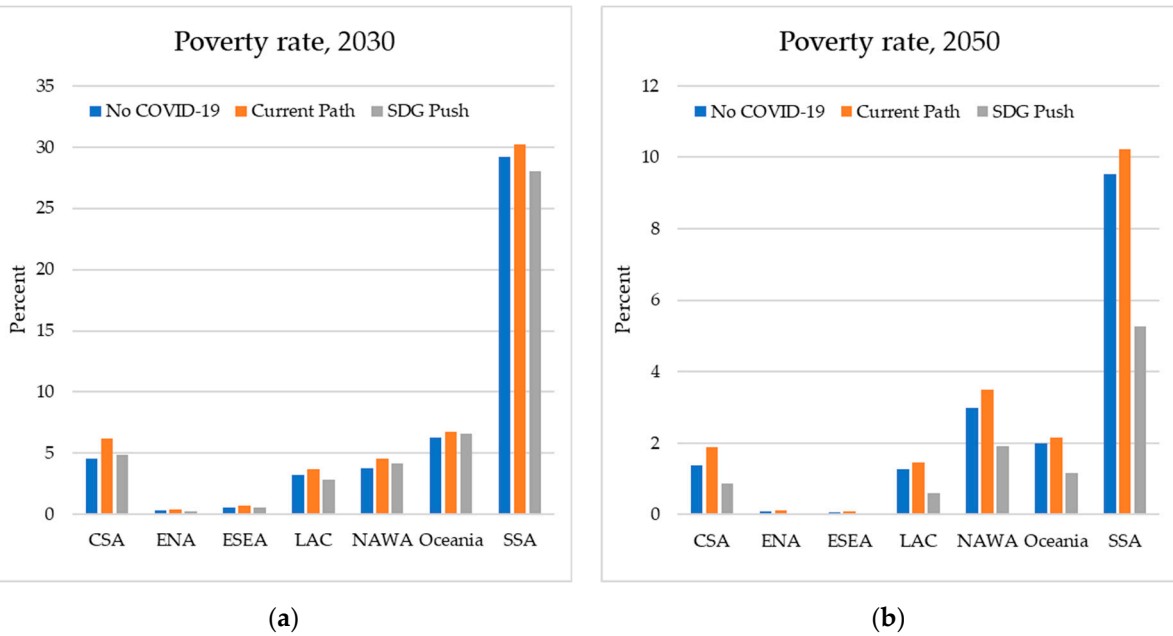

**Figure 3.** (**a**) Percent of population living on less than USD 2.15/day across scenarios by region in 2030. Source: IFs 8.10. (**b**) Percent of population living on less than USD 2.15/day across scenarios by region in 2050. Source: IFs 8.10.

By 2050, the *SDG Push* results in a significant decline in poverty rates across regions (Figure 3b). The SDG 1 goal of eliminating extreme poverty is achieved in all regions except SSA, where the poverty rate is still roughly half that projected along the *Current Path*.

There are various mechanisms in the SDG Push scenario that improve poverty outcomes relative to the Current Path. Government transfer programs boost incomes directly, while a number of interventions also work to alleviate poverty indirectly through improvements to the economy and human development. Family planning programs reduce the future investment required to achieve similar outcomes in areas of education and health, driving up human wellbeing and promoting productivity gains. Government spending is reoriented towards education, health, infrastructure, and R&D sectors, leading to greater long-term gains in multidimensional development. More efficient use of agricultural and energy resources also unlocks economic gains that facilitate reductions in poverty. This combination of direct and indirect interventions leads to a virtuous cycle towards the eradication of poverty.

### 3.2. Malnutrition

SDG 2 is to "End hunger, achieve food security and improved nutrition and promote sustainable agriculture" and targets range from ensuring food access for vulnerable populations to measures addressing agricultural investments and trade. For this analysis, we focus more narrowly on population-wide undernutrition. Full global results are available in Table 3.

Prior to the outbreak of COVID-19, we estimate that just under 8 percent of the global population (612 million people) suffered from malnutrition and that 67 countries had already met the SDG 2.1 goal of Zero Hunger. In a *No COVID-19* world, we project that malnutrition would continue to fall but would not achieve the goal at a global level. By 2030, still more than 5 percent of the population (445 million) would suffer from malnutrition, with 95 countries meeting the target of 3 percent. At a global level, the target would be

achieved by 2044, and by midcentury, the malnourished portion of the population would fall to 2.1 percent (203 million people).

**Table 3.** Results by scenario for SDG 2.1, using the percent of the population suffering from malnutrition. Source: IFs 8.10.

| Scenario | 2019 Global Value | 2019 Countries Meeting Target | 2030 Global Value | 2030 Countries Meeting Target | 2050 Global Value | 2050 Countries Meeting Target |
|---|---|---|---|---|---|---|
| *No COVID-19* | 7.9 | 67 | 5.3 | 95 | 2.1 | 134 |
| *Current Path* | 7.9 | 67 | 5.4 | 89 | 2.2 | 133 |
| *SDG Push* | 7.9 | 67 | 4.3 | 106 | 0.8 | 164 |

The COVID-19 outbreak in 2020 reduced economic growth globally and increased both poverty and hunger. We estimate that in 2020, the rate of malnutrition increased by nearly 0.5 percentage points or 37 million people relative to a *No COVID-19* scenario (Figure 4). As the world recovered from that initial shock, hunger began to fall again but remained higher than it would have been otherwise. By 2030, 15 million more people are projected to be malnourished in the *Current Path* scenario compared to a *No COVID-19* world. By 2050, 6.6 million more people are still projected to suffer from malnutrition as a result of the shadow of the pandemic.

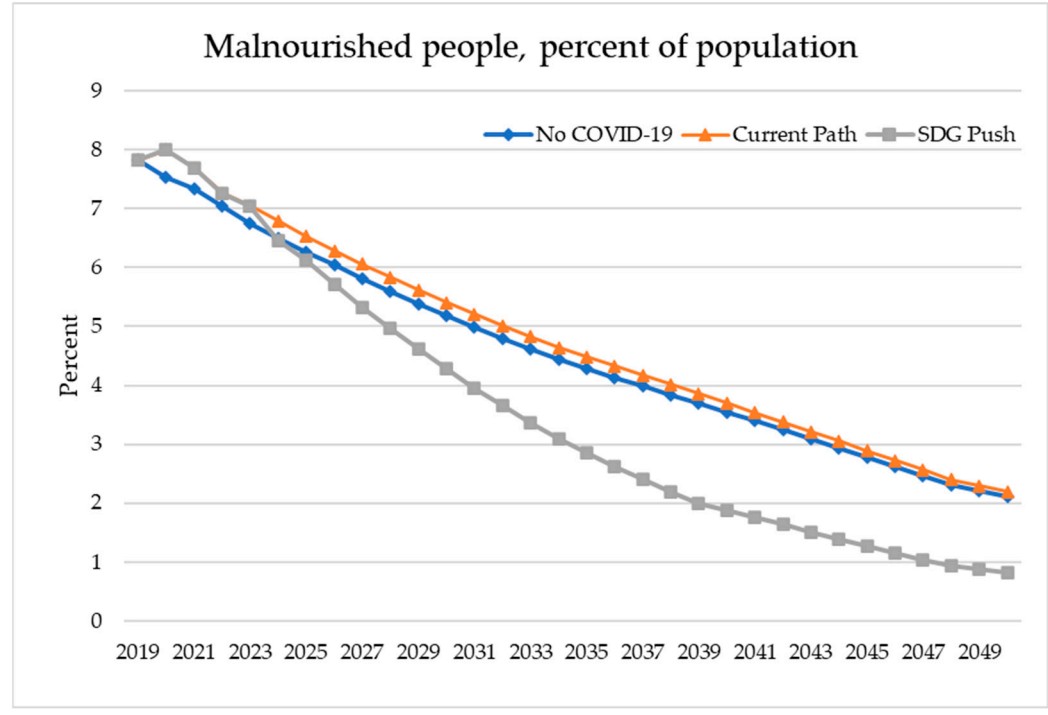

**Figure 4.** Percent of population malnourished, across the world, across scenarios. Source: IFs 8.10.

However, in an *SDG Push* world, multiple interventions are made to address hunger through both food supply and accessibility. By 2030, global malnutrition is reduced by 1.1 percentage points compared to the *Current Path*, and by 2035 the global malnutrition rate falls below 3 percent, ten years before it is projected to in the *Current Path*. By 2050, the malnourished population falls to 0.8 percent (77 million) and 164 countries have met the SDG 2.1 target—31 more than are projected to do so in the *Current Path*.

As in poverty, CSA is the region that experienced the largest increase in malnourishment due to COVID-19 (Figure 5). In 2020, the *Current Path* reflects an additional 22 million

people in the region pushed into malnutrition compared with a *No COVID-19* scenario (Figure 5), followed by SSA with just over 6 million. By 2050, the COVID-19 effect is not as large but still at nearly 5 million malnourished people, while the effect in SSA falls to meet that of many other regions. However, the effect also remains significant in NAWA, where over 2 million more people remain malnourished in 2050 in the *Current Path*.

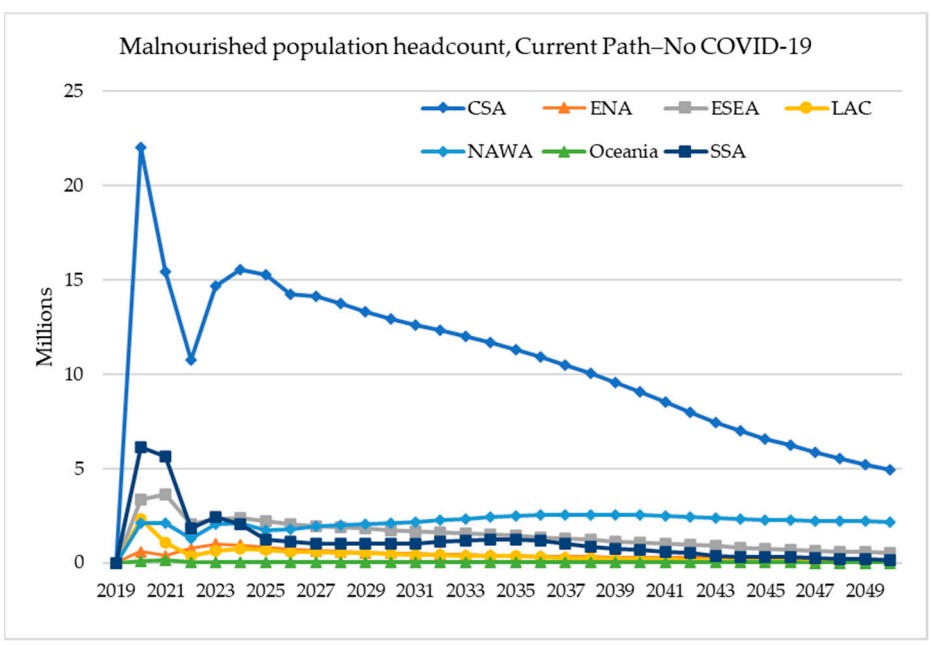

**Figure 5.** Difference between the number of people with malnutrition in the *Current Path* scenario and the *No COVID-19* scenario, by region. Source: IFs 8.10.

The *SDG Push* scenario simulates a gradual increase in equality of access to calories among other interventions. Even by 2030, the *SDG Push* results in a reduction in the rate of malnutrition below that in the *No COVID-19* scenario in all regions (Figure 6a).

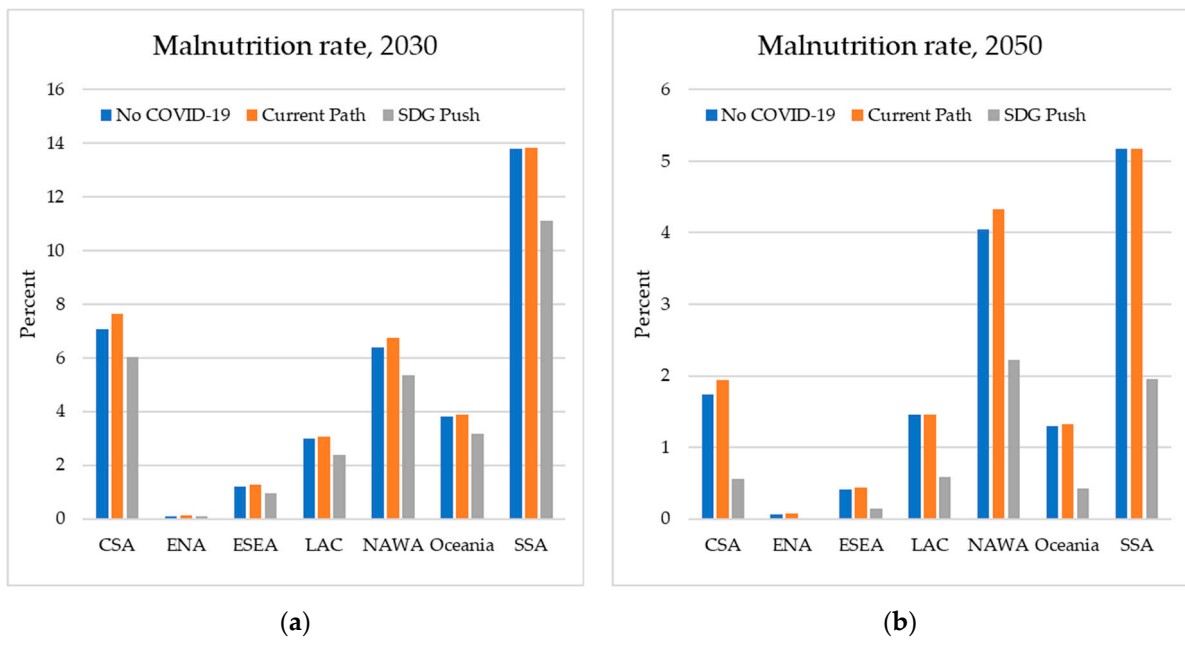

(**a**)                                                                                          (**b**)

**Figure 6.** (**a**) Percent of population suffering from malnutrition across scenarios by region in 2030. Source: IFs 8.10. (**b**) Percent of population suffering from malnutrition scenarios by region in 2050. Source: IFs 8.10.

Figure 6b shows the same results by region for 2050. Both the NAWA and SSA regions are projected to still fail to achieve the SDG 2 goal of eliminating hunger by the middle of the century in both the *Current Path* and the *No COVID-19* scenario. In the *SDG Push*, the target is met in all regions, and the rate of malnutrition relative to the *Current Path* is cut by between one-half and two-thirds.

The mechanism at play driving this developmental indicator down in the *SDG Push* scenario relative to the *Current Path* is primarily within the agricultural sector, which sees advances in production and yield but also fewer losses from future climate change (as the *SDG Push* scenario reduces carbon emissions relative to the *Current Path*). In addition, a powerful policy strategy to also reduce malnutrition improves the equality in the distribution of caloric resources in a society, a proxy intervention for programs that target food insecurity in the most poor and vulnerable populations.

*3.3. HDI*

While the HDI itself is not an SDG indicator, it reflects progression toward multiple SDGs directly (including SDG 3: Good Health and Wellbeing, SDG 4: Quality Education, and SDG 8: Decent Work and Economic Growth) and even more indirectly, as improving human development is associated with a myriad of other improvements, from poverty reduction to infrastructure access. Rather than projecting the index itself, IFs constructs the HDI each year from the projection of constituent parts defined by the UNDP: a long and healthy life (life expectancy), being knowledgeable (educational attainment and school life expectancy), and having a decent standard of living (GDP per capita).

In 2019, we estimate the global HDI at 0.710, roughly the level of Jamaica (0.710) or Botswana (0.707) and roughly 20 percent higher than India in the same year. In a *No COVID-19* scenario, this is projected to improve but gradually, reaching 0.739 by 2030 and 0.790 by 2050 (Table 4).

**Table 4.** Human Development Index (HDI) for the world across scenarios. Source: IFs. 8.10.

| Scenario | 2019 | 2030 | 2050 |
| --- | --- | --- | --- |
| *No COVID-19* | 0.710 | 0.739 | 0.790 |
| *Current Path* | 0.710 | 0.735 | 0.786 |
| *SDG Push* | 0.710 | 0.741 | 0.814 |

Driven primarily by the reduction in economic growth, the HDI fell by 0.002 in 2020 before beginning a gradual recovery with a forecast trajectory parallel to that of the *No COVID-19* scenario. By 2030, we project that the HDI in the *Current Path* would rise to 0.735 and by 2050 reach 0.786, nearly half a percent below the counterfactual in both years (Figure 7).

Through addressing barriers to growth, health, and education, the *SDG Push* accelerates improvements in human development and surpasses the *No COVID-19* trajectory by 2029. By 2030, in this scenario, the HDI reaches 0.741 and by 2050 0.814, a 3.5 percent improvement over the *Current Path*. While these improvements are relatively more limited than for poverty and hunger, the HDI is made up of components which change very slowly, such as life expectancy and adult educational attainment.

At a regional level, the CSA region experienced the greatest reduction in the HDI in the COVID-19 scenario relative to the *No COVID-19* scenario (Figure 8). This is not unexpected, as the region also suffered the greatest impact in terms of poverty and undernourishment, shown above. The ESEA region, on the other hand, experienced only a moderate reduction in the HDI in 2020 but is expected to be affected to a greater degree in the long run due to greater impacts on population life expectancy and educational attainment. Even so, the effect of COVID-19 across all regions is not large. Other than GDP per capita, the components of the HDI are measures of long-term stocks reflecting human health and education and should be expected to change very slowly.

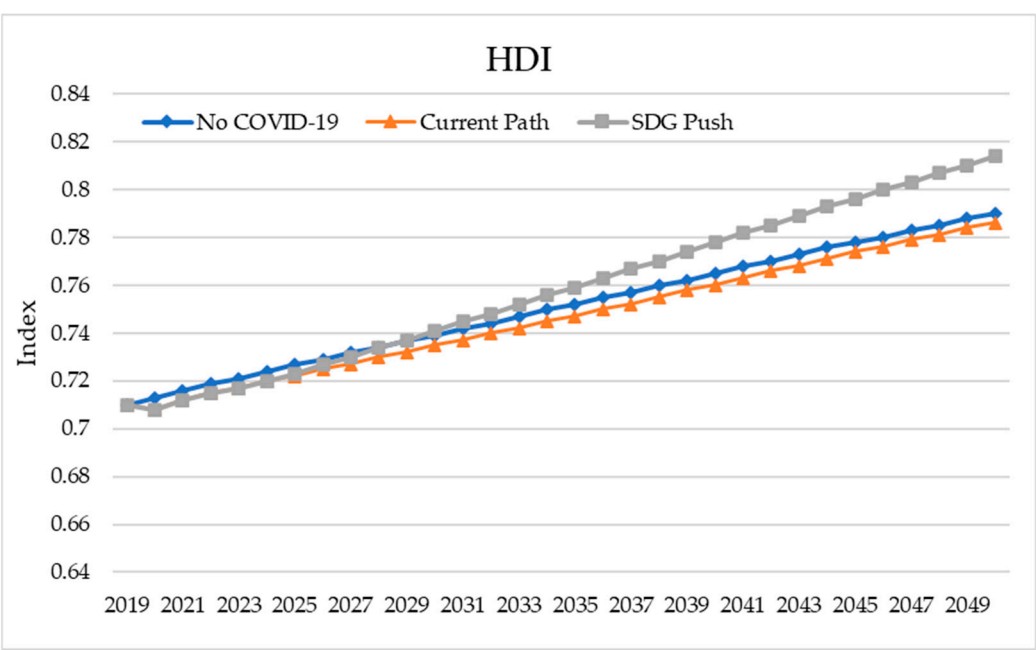

**Figure 7.** Human Development Index (HDI) across scenarios. Source: IFs. 8.10.

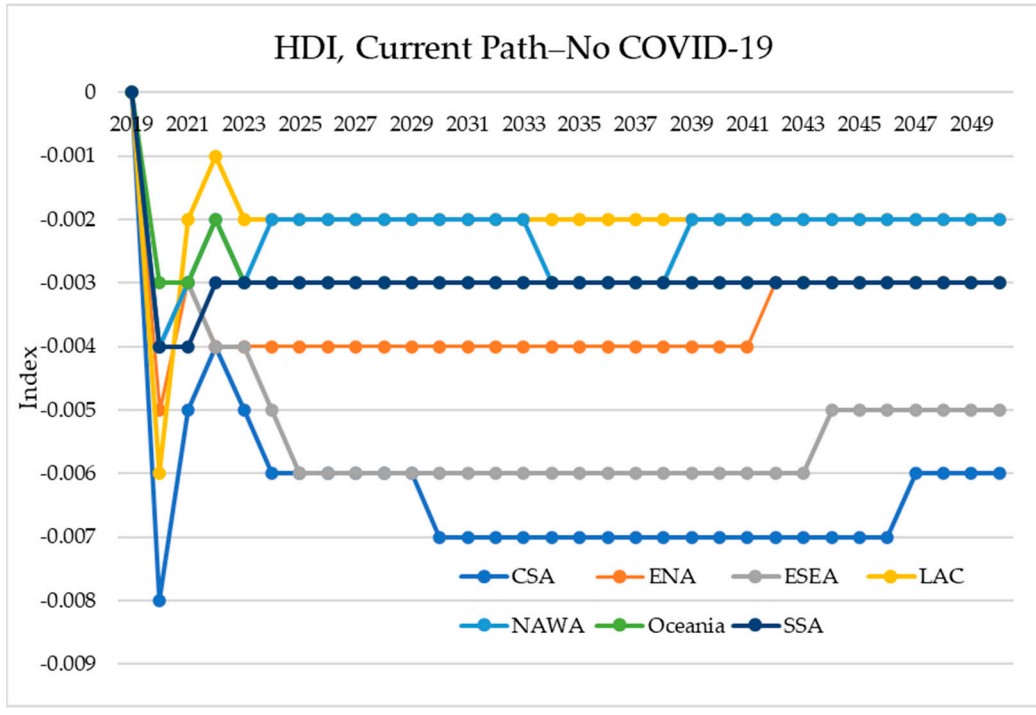

**Figure 8.** Difference between the HDI in the *Current Path* scenario and the *No COVID-19* scenario, by region. Source: IFs 8.10.

Figure 9 shows the effect of the *SDG Push* scenario on the HDI relative to the *Current Path* scenario by region. Here, the HDI values improve across all scenarios, and the effect of the *SDG Push* is significantly greater than the effect of COVID-19 even in very early years.

The greatest improvements are seen in regions where the HDI values are relatively low to begin with: SSA, CSA, and NAWA. But improvement is still seen in ENA, the region with the highest HDI even prior to COVID-19.

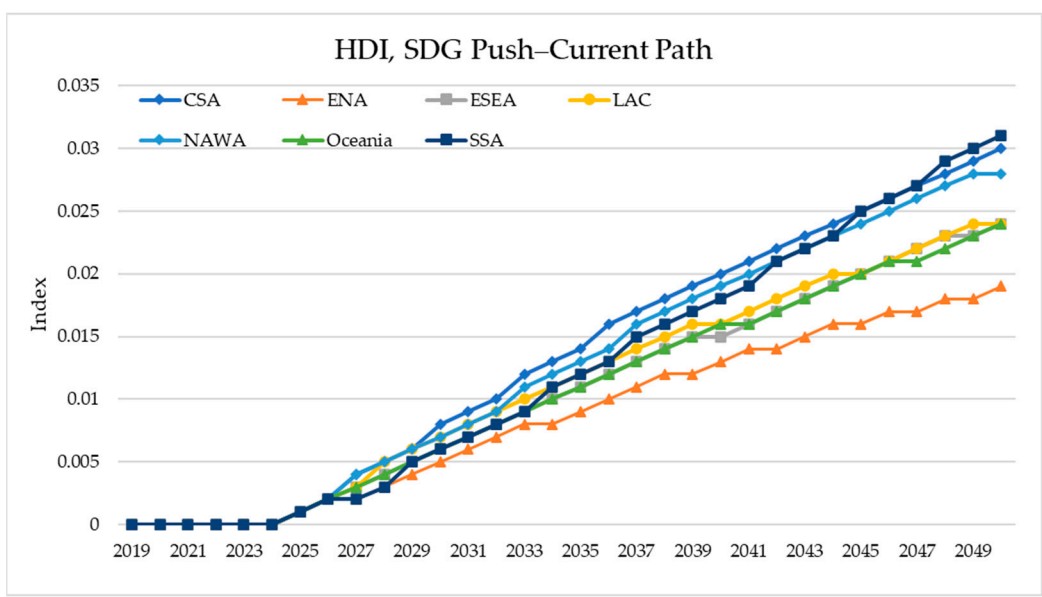

**Figure 9.** Difference between the HDI in the *Current Path* scenario and the *SDG Push* scenario, by region. Source: IFs 8.10.

Each of the policy strategies simulated in the *SDG Push* scenario leads to improvements in the HDI. Education policies increase attainment and school survival, direct health investments and indirect improvements in the proximate drivers of health outcomes (like improved access to water and sanitation) increase life expectancy, and the broad-based investments in multidimensional development increase productivity and raise GDP per capita.

## 4. Discussion

Even before the COVID-19 outbreak, we were not on a path to achieve the SDG targets assessed in this report, including eliminating poverty and hunger, as well as improving health and education, by the target date of 2030. This study seeks to (1) better understand how the COVID-19 pandemic has and will continue to set back progress toward the SDGs in the long run and (2) explore how an integrated development push could make up for this lost progress and accelerate development.

We find that the outbreak of COVID-19 and resulting containment measures, such as lockdowns and quarantines, did not have a devastating long-term effect on SDG achievement, though they resulted in further setbacks. The number of countries meeting the 2030 targets reduced from 113 to 107 for poverty and from 95 to 89 for malnutrition due to the COVID-19 shock. COVID-19 increased the global extreme poverty rate by 1 percentage point, and while its effect will soften over time, by 2030 (2050), 57 million (29 million) more people will live in extreme poverty compared with a *No COVID-19* scenario. Some regions have and will continue to recover more rapidly than others. A less severe economic effect and more rapid recovery in SSA in particular led to fewer people in the region pushed into poverty than previous research estimated [14,43,45]. On the other hand, the CSA region bore the brunt of the poverty impact of COVID-19, with more than twice as many people pushed into poverty than in SSA. Still, without additional action, no region is projected to catch up with its *No COVID-19* trajectory.

This research also shows that there is significant room for development improvement using an integrated approach to addressing key human and economic development deficiencies. The *SDG Push* scenario simulates the results of moving beyond recovery from a global crisis, demonstrating how an ambitious global push across issue areas can accelerate progress toward global development goals. This scenario closes the gap between the *Current Path* and the *No COVID-19* world in a matter of years and makes additional

gains toward SDG achievement. By 2050, 20 more countries achieve the SDG 1.1 target and 31 more countries achieve the SDG 2.1 target than expected in the *Current Path*. In this world, the goal of Zero Hunger is projected to be met just a few years behind the target date of 2030 and the goal of ending poverty a little more than a decade behind.

In comparing our results with the results of others, we note that there is general alignment that the COVID-19 pandemic has reduced human development outcomes and our ability to achieve the SDGs without additional policy interventions [5–8,10]. While we are unable to directly quantify what the magnitude of this effect is across SDGs in a composite way (as, say, the work of Li et al. [6]), we see a generally similar magnitude in these results compared with other work. Estimates for the immediate effect on poverty fall within the range provided by previous research [16,43–45]. While we are not aware of other studies which estimate the long-term effect of the pandemic on population-wide undernutrition, our findings are in line with previous work finding that the pandemic increased food unaffordability [17] and child undernutrition [19] in the short run and food insecurity [21] in the long run. These findings are also in line with research emphasizing the benefits of an integrated development push in accelerating progress toward SDG achievement [22].

From a methodological perspective, this work highlights the ongoing importance of representing governmental systems within integrated modeling frameworks to capture patterns of political choice that are crucial for thinking about the future socioeconomic development future on a planet characterized by finite resources. There are ongoing debates about the role of economic growth and technology as a way to balance human development and environmental systems [28,46,47]. More explicitly representing these processes in global modeling efforts will allow us to examine how governmental policies related to education, health, infrastructure, the military, and R&D can inform development in technology while also reducing unnecessary waste.

It is important to note that there are limitations to this work. First, the biggest difference between the *No COVID-19* and *Current Path* scenarios in this analysis comes from the difference in GDP growth rates as projected in 2019, prior to the outbreak, and more recent growth rate data and projections from 2023. In the initial years, COVID-19 can be assumed to have been the primary driver for most changes to growth rates and thus responsible for most of the economic and development effects. In more recent years, additional shocks both local and global have occurred that have also likely changed economic growth trajectories, including but not limited to the onset of wars in Ukraine and more recently in Gaza. This analysis does not seek to isolate the effect of the COVID-19 pandemic from these other shocks but to contrast the *Current Path* we are on today with the trajectory the world was on prior to the outbreak.

Another limitation to the assessment of the effect of the pandemic is the lack of sector-specific effects modeled. COVID-19 had different effects across economic sectors, including hitting some sectors, like tourism and transport, particularly hard. These sectors are not differentiated in IFs and thus may not be fully accounted for here.

Yet another limitation of this work is our inability to effectively treat uncertainty in the analysis. First, the scenarios that are presented here do not include traditional longitudinal "confidence intervals" because the structure of the model and the large number of interconnected systems makes this kind of uncertainty framing implausible. Second, the scenarios themselves are not attempts to account for the actual uncertainty inherent in the global system—as a global pandemic suggests, future human development will be impacted by a wide range of exogenous factors, and it is beyond the scope of this work to extrapolate here. Finally, as all models are representations of reality that make significant simplifying assumptions, the use of tools for planning should be understood to be illustrative and exploratory, not predictive and prescriptive.

This analysis is focused largely on a select few human-development-oriented goals and targets that are most relevant for low- and lower-middle-income countries. This is just one component of the entire SDG agenda and does not address countries that have, for

example, eliminated extreme poverty but still have poverty at higher levels. In focusing on these SDGs and indicators, we focus on populations that are especially vulnerable and addressing core development needs. Moreover, these goals have strong connections with many others—for instance, eradicating poverty has synergies with many other SDGs [48], while failing to meet SDGs 1 and 2 would seriously undermine the ability to meet many others [49,50].

Finally, this analysis also does not address the many sustainability- and environmentally oriented SDGs, like SDGs 13, 14, and 15. Some of the literature indicates that lockdown and quarantine measures, especially reducing road and air travel, had positive effects on reducing emissions and meeting some environmental SDG targets [51,52]. However, much of this was temporary. And the pause in growth and investment could have adverse long-term consequences on environmental progress, outweighing the initial benefits [51].

As the world faces future global shocks and challenges, it is vital to better understand both how those shocks may impact development in the long run and how policy effort can support recovery from and even beyond those impacts. We find that for a shock like that imposed by the COVID-19 pandemic, the consequences to human wellbeing are real, but transformative and integrated policy interventions can lead to much greater benefits.

**Supplementary Materials:** The following supporting information can be downloaded at: https://www.mdpi.com/article/10.3390/su16083309/s1, Table S1: Detailed interventions for the SDG Push scenario.

**Author Contributions:** Conceptualization, B.B.H., J.D.M., B.A. and L.P.; Methodology and Data Support, D.K.B., M.T.I. and J.S.; Formal Analysis, B.B.H., T.H., D.K.B and J.D.M.; Writing—Original Draft Preparation, T.H.; Writing—Review and Editing, J.D.M., B.B.H. and M.T.I. All authors have read and agreed to the published version of this manuscript.

**Funding:** This research was partly funded by the United Nations Development Programme.

**Institutional Review Board Statement:** Not applicable.

**Informed Consent Statement:** Not applicable.

**Data Availability Statement:** Data used and full results can be accessed and replicated by downloading the IFs model version 8.10, located at the following URL: https://ifs02.du.edu/IFs%20with%20Pardee%208_10%20SDG%20Push%20Rev%20February%2022%202023.zip (accessed on 22 February 2024).

**Acknowledgments:** The authors would like to thank the following individuals for their support in this research process: Kaylin McNeil for early research support that contributed to this work, Yutang Xiong for data administration and support of the IFs model, Pam Hoberman for administrative and project management support, and Joanna Felix and Serge Kapto for involvement in conceptualization and analysis of earlier stages of the scenarios featured here.

**Conflicts of Interest:** Authors Babatunde Abidoye and Laurel Patterson are employed by funding organization UNDP and had a role in the early conceptualization and design of this study. The remaining authors declare no conflicts of interest.

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
