# Peer review of "Sustainable Development Goal Attainment in the Wake of COVID-19: Simulating an Ambitious Policy Push"

_sustainability, doi:10.3390/su16083309_

Round 1

Reviewer 1 Report

Comments and Suggestions for Authors

This is a well-written paper on a relevant topic. The paper clearly shows the challenges that emerged from the COVID pandemic with respect to achievement of SDGs related to poverty and hunger. 

There are three issues I want to raise:

1.        The surrounding text in the Results section adds to what is presented in the tables and graphs. There are some interesting issues that could be further discussed, especially with respect to the SDG push scenario (see next point).

2.        The SDG push scenario provides great HWB outcomes, but how is largely a black box. The SI presents assumptions in very technical terms that only an IFs modeler would understand. Best to put a simplification of the table of the SI and some related text in section 2.2.3 to allow the reader to better understand what is happing in this scenario. The Results section should then use this background to better explain the results. What are the key strategies? Are they different for poverty, hunger and overall development? And what is additionally needed due to set-backs resulting from the pandemic?

3.        A discussion of the uncertainties related to projections is largely missing. The results section could devote some text to these uncertainties and how they could change the outcomes and conclusions (or not). Furthermore, although the authors range some issues with earlier projections, I am still interested how some of these relate to your modelling results.

Assuming the authors would somehow address these comments, I am looking forward to seeing this paper published in Sustainability.

Author Response

Reviewer 1:  This is a well-written paper on a relevant topic. The paper clearly shows the challenges that emerged from the COVID pandemic with respect to achievement of SDGs related to poverty and hunger. 

The surrounding text in the Results section adds to what is presented in the tables and graphs. There are some interesting issues that could be further discussed, especially with respect to the SDG push scenario (see next point).

Author response:  We significantly expanded our treatment of the SDG Push scenario and connected it more carefully to the results section as well as core assumptions in the IFs model.  We also expanded our discussion section to include consideration of these results as well.

Reviewer 1:  The SDG push scenario provides great HWB outcomes, but how is largely a black box. The SI presents assumptions in very technical terms that only an IFs modeler would understand. Best to put a simplification of the table of the SI and some related text in section 2.2.3 to allow the reader to better understand what is happing in this scenario. The Results section should then use this background to better explain the results. What are the key strategies? Are they different for poverty, hunger and overall development? And what is additionally needed due to set-backs resulting from the pandemic?

Author response:  We thank you for this very helpful comment and significantly expanded our description of the scenario assumptions in the SDG Push intervention.  We also added paragraphs at the end of each results section that connected the results with the SDG Push interventions via the IFs modeling framework logic.  The idea here is to better communicate the key strategies that can be pursued for each of the indicators we mention and to simplify the material in the SI.

Reviewer 1:  A discussion of the uncertainties related to projections is largely missing. The results section could devote some text to these uncertainties and how they could change the outcomes and conclusions (or not). Furthermore, although the authors range some issues with earlier projections, I am still interested how some of these relate to your modelling results.

Author response:  We included a more forceful discussion of uncertainties in the discussion section.  This highlights three ways in which we can think about uncertainty using a modeling framework like IFs.

Reviewer 2 Report

Comments and Suggestions for Authors

This article assesses the impact of the COVID-19 pandemic on progress towards the SDGs and examines strategies for recovery and accelerated development. This manuscript helps to understand how the COVID-19 pandemic has affected progress towards achieving Sustainable Development Goal 1 No Poverty, Sustainable Development Goal 2 Zero Hunger, and improvements in the Human Development Index (HDI), which summarises progress towards several of the SDGs.

This article is relevant because it contributes to a comprehensive understanding of the pandemic's broad-ranging effects on global development objectives. The interconnectedness of the examined SDGs and the HDI underscores the complex and multifaceted nature of the pandemic's impact on societal well-being. The Authors argue that, from a methodological point of view, this paper highlights the ongoing importance of representing governmental systems within integrated modeling frameworks to capture patterns of political choice that are crucial for thinking about the future socioeconomic development future on a planet characterized by finite resources.

The idea of conducting the study in three scenarios is commendable: a scenario that represents the world before the COVID-19 outbreak; a scenario that reflects a baseline future development path, including the impact of COVID-19; and a scenario that simulates an integrated movement towards the SDGs through global interventions.

The study's three-scenario approach is crucial as it provides a comprehensive analysis: comparing pre-COVID conditions, projecting realistic outcomes with pandemic impacts, and simulating an integrated pursuit of Sustainable Development Goals.

The paper exhaustively presents and justifies the materials and methods used in the study.

The conclusions contain limitations of the study, according to which there is a need to take into account additional local and global shocks, as well as modelling of sector-specific effects.

Author Response

Reviewer 2: This article assesses the impact of the COVID-19 pandemic on progress towards the SDGs and examines strategies for recovery and accelerated development. This manuscript helps to understand how the COVID-19 pandemic has affected progress towards achieving Sustainable Development Goal 1 No Poverty, Sustainable Development Goal 2 Zero Hunger, and improvements in the Human Development Index (HDI), which summarises progress towards several of the SDGs.

This article is relevant because it contributes to a comprehensive understanding of the pandemic's broad-ranging effects on global development objectives. The interconnectedness of the examined SDGs and the HDI underscores the complex and multifaceted nature of the pandemic's impact on societal well-being. The Authors argue that, from a methodological point of view, this paper highlights the ongoing importance of representing governmental systems within integrated modeling frameworks to capture patterns of political choice that are crucial for thinking about the future socioeconomic development future on a planet characterized by finite resources.

The idea of conducting the study in three scenarios is commendable: a scenario that represents the world before the COVID-19 outbreak; a scenario that reflects a baseline future development path, including the impact of COVID-19; and a scenario that simulates an integrated movement towards the SDGs through global interventions.

The study's three-scenario approach is crucial as it provides a comprehensive analysis: comparing pre-COVID conditions, projecting realistic outcomes with pandemic impacts, and simulating an integrated pursuit of Sustainable Development Goals.

The paper exhaustively presents and justifies the materials and methods used in the study.

The conclusions contain limitations of the study, according to which there is a need to take into account additional local and global shocks, as well as modelling of sector-specific effects.

Author response:  Thank you.

Reviewer 3 Report

Comments and Suggestions for Authors

Dear authors,

The introduction of the article is highly promising, providing insight into the research issue—Sustainable Development Goals and the impacts of Covid-19 on their attainment. The authors should emphasize the identified gap in the literature more clearly and articulate the research's purpose and objectives. Following this, hypotheses or research questions should be formulated. Additionally, at the conclusion of the introduction, the authors are encouraged to outline the paper's structure. It is important to note that the results of the research or their contribution to advancing the literature should not be presented in the introductory section. After the introduction, the authors are required to conduct a literature review, which is currently conspicuously missing. Therefore, they need to include a new section before the materials and methods, a comprehensive literature study that incorporates relevant citations related to the researched issue.

The material and method section is very well written and needs no improvement. The results are also clearly presented. 

In a scientific article, the discussion section is a critical component where the authors interpret and analyze the results presented in the earlier sections, such as the methodology and results. The purpose of the discussion is to provide context to the findings, explore their implications, compare them with existing literature, and draw conclusions.

Hence, a significant shortcoming in the discussion section is the absence of a comparison between its own results and those of previous studies. A thorough discussion typically includes a comparison of the current findings with those of earlier studies, allowing authors to highlight similarities, differences, or advancements in knowledge.

Every scientific article concludes with a conclusion section, and it is currently missing in this work. The authors will need to formulate the conclusion section to effectively summarize and wrap up the key findings and implications of their research.

Author Response

Author Response:  While we agree that hypothesis oriented research is indeed important, it is often not an approach deployed by researchers working with integrated modeling frameworks.  We would argue that there are various ways to build scientific knowledge and that hypothesis testing is indeed useful, but exploratory scenario analysis and integrated modeling approaches can provide other kinds of information that are also useful, such as evaluating the magnitude and direction of trends, accounting for a broad set of changing factors that dynamically change all aspects of the modeled landscape, etc. 

Reviewer 3:  Additionally, at the conclusion of the introduction, the authors are encouraged to outline the paper's structure.

Author response:  Thank you—we added a paragraph to the end of the introduction and believe this strengthens the presentation of work.

Reviewer 3: It is important to note that the results of the research or their contribution to advancing the literature should not be presented in the introductory section. After the introduction, the authors are required to conduct a literature review, which is currently conspicuously missing. Therefore, they need to include a new section before the materials and methods, a comprehensive literature study that incorporates relevant citations related to the researched issue.

Author response:  We thank the reviewer for this helpful comment and agree that literature reviews are an important aspect of scientific work.  To that end, we did extend the number of citations we included in our overview of the attempts to quantify the future impact of COVID on development (see our comment above).  However, we would respectfully push back about the need to include a separate section on literature.  While this is indeed a very common practice, we guided the development of our response to the journal by using the template that Sustainability provides for creating and submitting a manuscript which does not have a literature section.  See here:  https://www.mdpi.com/files/word-templates/sustainability-template.dot

Reviewer 3: The material and method section is very well written and needs no improvement. The results are also clearly presented. 

Author response:  We thank you kindly for this feedback.

Reviewer 3: In a scientific article, the discussion section is a critical component where the authors interpret and analyze the results presented in the earlier sections, such as the methodology and results. The purpose of the discussion is to provide context to the findings, explore their implications, compare them with existing literature, and draw conclusions.

Hence, a significant shortcoming in the discussion section is the absence of a comparison between its own results and those of previous studies. A thorough discussion typically includes a comparison of the current findings with those of earlier studies, allowing authors to highlight similarities, differences, or advancements in knowledge.

Author response:  We extended the discussion to include a more explicit comparison with previous literature.

Reviewer 3: Every scientific article concludes with a conclusion section, and it is currently missing in this work. The authors will need to formulate the conclusion section to effectively summarize and wrap up the key findings and implications of their research.

Author response:  We also followed the Sustainability template for guiding our decision about the inclusion of a Conclusion section.  See here:  https://www.mdpi.com/files/word-templates/sustainability-template.dot  In that document, they refer to the Conclusion section with the following language:  “This section is not mandatory but can be added to the manuscript if the discussion is unusually long or complex.”

Round 2

Reviewer 3 Report

Comments and Suggestions for Authors

The paper has been significantly improved. However, the discussion section still does not meet the standards of a scientific paper. Improving the discussion section of a scientific article involves several key steps: summarize key findings, relate findings to research questions, compare with previous research.

Author Response

Response: Thank you for this comment. We have revisited the discussion section in an attempt to more explicitly and methodically draw attention to the important components highlighted by: reiterating the core research questions (section paragraph 1), expanding the summary findings (paragraphs 2-4) pertaining to those research questions, and expanding the findings comparison to compare this research with findings across the research questions (paragraph 5).